



# An adaptive unstructured mesh based solution to topography least-squares reverse-time imaging

Qiancheng Liu[1] and Jianfeng Zhang[2]

[1]Department of Physical Sciences and Engineering, King Abdullah University of Science and Technology
[2]Institute of Geology and Geophysics, Chinese Academy of Sciences

**Correspondence:** qiancheng.liu@kaust.edu.sa

**Abstract.** Least-squares reverse-time migration (LSRTM) attempts to invert for the broadband-wavenumber reflectivity image by minimizing the residual between observed and predicted seismograms via linearized inversion. However, rugged topography poses a challenge in front of LSRTM. To tackle this issue, we present an unstructured mesh-based solution to topography LSRTM. As to the forward/adjoint modeling operators in LSRTM, we take a so-called unstructured mesh-based "grid method".

Before solving the two-way wave equation with the grid method, we prepare for it a velocity-adaptive unstructured mesh using a Delaunay Triangulation plus Centroidal Voronoi Tessellation (DT-CVT) algorithm. The rugged topography acts as constraint boundaries during mesh generation. Then, by using the adjoint method, we put the observed seismograms to the receivers on the topography for backward propagation to produce the gradient through the cross-correlation imaging condition. We seek the inverted image using the conjugate gradient method during linearized inversion to linearly reduce the data misfit function.

Through the 2D SEG Foothill synthetic dataset, we see that our method can handle the LSRTM from rugged topography.

*Copyright statement.* The copyright notice will be updated ...

## 1   Introduction

Seismic imaging is a geophysical technique that investigates the subsurface using observed sesimograms. Rugged topography is an important issue in exploration geophysics, which is usually present with low-velocity weathered layers, high-velocity

outcrops of steep dips. Conventional static corrections (Shtivelman and Canning, 1988) may produce imperfect imaging results when complicated geological structures with rugged topography occur. In this case, reverse-time migration (RTM) (Whitmore, 1983) comes into being a proper solution.

   Based on the two-way wave equation, RTM has proven to be an important tool in imaging complex subsurface structures without dip limitation (Whitmore, 1983; Baysal et al., 1983). RTM has also got applications to ground-penetrating radar

imaging (Foroozan and Asif, 2010; Liu et al., 2017a), medical imaging (Kosmas and Rappaport, 2006; Wang et al., 2016) and waveform inversion (Van Leeuwen and Mulder, 2010; van Leeuwen and Herrmann, 2013; Van Leeuwen and Herrmann, 2013; Kadu et al., 2017). However, in practice, the survey zone usually has the topography effect (Burgin et al., 2014). Several papers




on the topography RTM applications have been published. McMechan and Chen (1990) produce a topography RTM image by injecting the traces onto the nodes at different elevations within the finite-difference stencil. Rajasekaran and McMechan (1995) further promote this scheme to real data. However, the regular finite difference stencil cannot get rid of the staircase approximation. To overcome this limitation, several approaches considering the topography effect are proposed. Lan et al. (2014) cope with the rugged topography by flatting it to create a curvilinear mesh. Shragge (2014) deals with RTM from topography with the generalized coordinate system. However, these two techniques involve a "flattening" strategy, which cannnot handle a very rugged topography. Liu et al. (2017b) propose to deal with rugged topography RTM with adaptive, unstructured triangular meshing.

RTM, however, still acts as the adjoint operator to the linearized forward operator (Born modeling) rather than the inverse operator (Claerbout, 1992), leading to imperfect images. To overcome this imperfectness, LSRTM attempts to approach the inverted operator via linearized inversion. The concept of least-squares migration (LSM) dates back to Lailly (1983). To date, LSRTM has got widespread attention and development. Wong et al. (2011) use LSRTM to image the ocean-bottom data. Dai and Schuster (2013) improve the computational efficiency of LSRTM by exploiting source-encoding. Zhang et al. (2015) propose a practical LSRTM solution by using a cross-correlation objective function. Yao and Jakubowicz (2016) formulate LSRTM in a generalized matrix form. Liu (2016) presents that eliminating the redundant effect of source wavelet in RTM can improve the convergence of LSRTM. Liu et al. (2017c) introduce a prestack cross-correlative LSRTM approach. Xu and Sacchi (2017) formulate the exact adjoint operator to the forward modeling to improve the performance of LSRTM. Liu and Peter (2018) introduce a one-step data-domain LSRTM.

In this paper, we explore to handle LSRTM from rugged topography by using a modeling solver named "grid method" (Zhang and Liu, 1999; Zhang, 2004; Gao and Zhang, 2006) based on an adaptive unstructured mesh. The computational cost and the stability of the "grid method" are similar with those of finite-difference method at $O(dt^2, dx^2)$ (Zhang and Liu, 1999). The unstructured mesh, which is velocity-adaptive, is generated before wavefield simulation. During the mesh generation, the rugged topography serves as boundary constraints. The meshing is based on the Delaunay Triangulation plus Centroid Voronoi Tessellation (DT-CVT) algorithm (Du et al., 1999; Du and Gunzburger, 2002; Du et al., 2006). The generated adaptive mesh has coarser and finer grids for higher and lower velocities, respectively. However, for a given modeling method, the adaptive unstructured mesh can help run the seismic simulation at a lower computational cost, which alleviates the computational burden of topography LSRTM. When the mesher and solver are ready, we begin to run the LSRTM workflow with MPI (Message Passing Interface) in parallel.

We organize the paper as follows. First, we review the grid method. Then, we present the adaptive, unstructured-mesh generation, with rugged topography as constraints. Next, we introduce the topography LSRTM workflow. Finally, we validate our methods on the Foothill model.




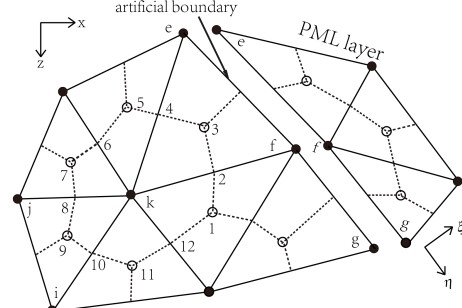

**Figure 1.** Local mesh of the grid method. The basic cells are in solid lines. The dashed lines link the centroids of the cells and the midpoint of the edges. The pressure is defined on the nodes in black dots, and the velocity and density at the center of cells in empty circles.

## 2 Theories and Methods

### 2.1 Solver: the grid method

As an unstructured mesh-based approach, the grid method can model seismic wave propagation in heterogeneous media by accurately considering the surface and interface topographies (Zhang and Liu, 1999; Zhang, 2004; Gao and Zhang, 2006). Let us start with the 2D acoustic wave equation

$$\frac{1}{c(x,z)^2}\frac{\partial^2 P(x,z,t)}{\partial t^2} = \nabla^2 P(x,z,t),$$ (1)

where $c(x,z)$ is the acoustic velocity and $P(x,z,t)$ the pressure. Fig. 1 illustrates the local mesh in the grid method. We assign the velocity parameters $c(x,z)$ at the centroids of the cells, and the pressure $P(x,z,t)$ at the nodes. The dash lines link the cell centroids and the edge midpoints. Given node $k$ centered in the domain $\Omega_k$ circles by contour 1-2-3-4-5-6-7-8-9-10-11-12-1, we integrate both sides of Eq. (1), yielding

$$\iint_{\Omega_k}\frac{1}{c^2}\frac{\partial^2 P}{\partial t^2}dxdz = \oint\left(\frac{\partial P}{\partial x}dz - \frac{\partial P}{\partial z}dx\right) = \sum_{l=1}^{m}\int_{S_{kl}}\frac{\partial P}{\partial x}dz - \sum_{l=1}^{m}\int_{S_{kl}}\frac{\partial P}{\partial z}dx,$$ (2)

with $S_{kl}$ being the curve of the $l$ cell around node $k$, and $m$ the number of cells around node $k$. For example, in Fig. 1, node $k$ is enclosed by contour 1-2-3-4-5-6-7-8-9-10-11-12-1 within its surrounding six cells.

After taking $M_k = \iint_{\Omega_k}1/c^2 dxdz$ inside domain $\Omega_k$, the left-hand side of Eq. (2) reduces to $M_k(\partial^2 P/\partial t^2)_k$. For a typical cell $ijk$, we calculate the first-order triangular-difference operators Cook et al. (2007) as

$$\frac{\partial P}{\partial x} = D_x P \approx \frac{1}{\Delta}\left[b_i(P)_i + b_j(P)_j + b_k(P)_k\right],$$
$$\frac{\partial P}{\partial z} = D_z P \approx \frac{1}{\Delta}\left[a_i(P)_i + a_j(P)_j + a_k(P)_k\right],$$ (3)




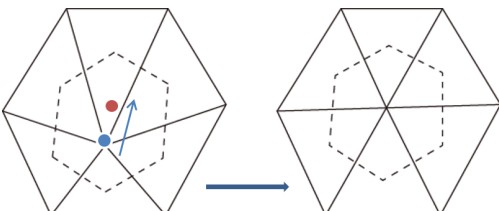

**Figure 2.** A toy model about how DT-CVT works. The Delaunay Tessellations are in solid lines and the Voronoi Tessellations in dashed lines. After the mesh optimisation by DT-CVT, we can see that the blue node moves to the position of the brown node. The optimised mesh on the right-hand-side has better mesh quality. That is, each cell approaches more to a regular triangle that those on the left-hand-side.

with $\Delta$ being the area of triangle $ijk$, $a_i = (x_k - x_j)/2, b_i = (z_k - z_j)/2$, and so on. Substituting Eq. (3) into the right-hand-side of Eq. (2) yields

$$\oint \left( \frac{\partial P}{\partial x} \alpha + \frac{\partial P}{\partial z} \beta \right) ds = \sum_{l=1}^{m} (D_x P)_l (b_k)_l - \sum_{l=1}^{m} (D_z P)_l (a_k)_l. \tag{4}$$

Then, we obtain a weak form of Eq. (1) represented around node $k$ as

$$5 \quad M_k \left( \frac{\partial^2 P}{\partial t^2} \right)_k = \sum_{l=1}^{m} (D_x P)_l (b_k)_l - \sum_{l=1}^{m} (D_z P)_l (a_k)_l. \tag{5}$$

With an $O\left(dt^2\right)$ approximation to $\left(\partial^2 P/\partial t^2\right)_k$, we update the pressure from $t$ and $t - dt$ to $t + dt$ using

$$(P)_k^{t+dt} = 2(P)_k^t - (P)_k^{t-dt} + dt^2 \left(\partial^2 P/\partial t^2\right)_k^t. \tag{6}$$

In numerical implementations, we calculate and store the parameters $M$, $a$, $b$ and $\Delta$ in advance. We only keep the pressures on the nodes, the number of which is twice smaller than that of the cells. In terms of the computational cost and the accuracy and stability, this method is similar with the $O(dt^2, dx^2)$ finite-difference method (Zhang and Liu, 1999).

### 2.2 Mesher: Delaunay Triangulation plus Centroid Voronoi Tessellation

The quality of unstructured mesh is crucial for the numerical simulation. We generate an adaptive mesh by using the Delaunay Triangulation (DT) plus Centroidal Voronoi Tessellation (CVT) (Du et al., 1999; Du and Gunzburger, 2002; Du et al., 2006) algorithm. In computational mathematics, the DT is a special triangulation for a given set of points. DT algorithm maximizes the minimum angle of all the triangle angles. This property can be used to avoid sliver triangles such that we choose DT for the generation of an initial mesh. The mesher honors the rugged topography by taking it as constraint boundaries. Then, we assign the generators onto the boundaries, with spatial intervals adaptive to the local velocities. Next, we circularly "pave" new Delaunay cells from the boundaries toward the interior with the doubly-linked data structure (Blacker and Stephenson, 1991). Note that a singly-linked list is insufficient for "circularly paving" because one cannot remove a singly-linked list entry in constant time.



However, the initial mesh lacks quality control (Du and Gunzburger, 2002), and its cell sizes change dramatically, as shown in Fig. 3a. Therefore, we attempt to perform mesh optimisation by using the Centroidal Voronoi Tessellation (CVT) algorithm. The Voronoi Tessellation of a set of points is dual to its Delaunay triangulation. CVTs are a special kind of special Voronoi Tessellations, whose tessellation generators are coincident with the "mass" centroids of the corresponding Voronoi regions

regarding a given "density" function (Du and Gunzburger, 2002). Because the spatial samplings are adaptive to the local migration velocities, in the 2D case, we define the "density" function $\sigma$ in CVTs as

$$\sigma\left(x,z\right)=\left(\frac{c_{\max}}{c_0\left(x,z\right)}\right)^2, \tag{7}$$

where $c_{\max}$ is the velocity maximum, and $c_0\left(x,z\right)$ the migration velocity. Given a Voronoi cell $V$, its "mass" centroid $(x^*,z^*)$ is represented as

$$x^*=\frac{\int_V x\sigma\left(x,z\right)dxdz}{\int_V \sigma\left(x,z\right)dxdz},z^*=\frac{\int_V z\sigma\left(x,z\right)dxdz}{\int_V \sigma\left(x,z\right)dxdz}. \tag{8}$$

For Voronoi tessellations $\{V_i\}_{i=1}^k$, we define the cost (energy) function concerning generators $\{x_i,z_i\}_{i=1}^k$ as

$$F\left(\{x_i,z_i\}_{i=1}^k\right)=$$
$$\sum_{i=1}^k \int_{V_i}\sigma\left(x,z\right)((x-x_i)^2+(z-z_i)^2)dxdz, \tag{9}$$

in which $F\left(\{x_i,z_i\}_{i=1}^k\right)$ can be treated as the summed rotational inertia regarding the velocity model. For minimization of Eq. (9), we apply an algorithm named Lloyd's method (Du et al., 1999; Du and Gunzburger, 2002), which iteratively

constructs the tessellations and centroids of the Voronois. Fig. 2 shows a toy model about how DT-CVT algorithm works. Note that while the interior generators are dynamically moving, we must keep the boundary control points fixed to maintain the topography information. With curved boundary constraints, Fig. 3 shows the comparison between the initial mesh (Fig. 3a) and the optimised mesh (Fig. 3b). With each cell approximating a normal triangle we can see that the latter has better mesh quality over the former. Compared with simpler meshing techniques (for example, only by DT), our technique performs well

in controlling the mesh quality.

The unstructured mesh has advantage in depicting irregular interfaces. The unstructured mesh of optimised quality benefits the seismic modeling in terms of accuracy and computational cost. For example, given the initial mesh in Fig. 3a by a simpler mesher, we can see there are some skewed cells inside. In seismic modeling, the time sampling interval is determined by the smallest space interval. According to our experiences in inhomogeneous (also as shown in Fig. 3b), the smallest interval in

an optimised mesh is twice larger than that in a simpler unstructured mesh, which helps alleviate the expensive computational cost of the seismic modeling in LSRTM.

The engine of LSRTM is the two-way wave-equation based seismic modeling. The accuracy together with flexibility for seismic modeling is up to its mesh discretization, especially for irregular strong discontinuities. The adaptive unstructured mesh has advantage in accurately depicting discontinuous interfaces. To validate this capability, we compare our unstructured



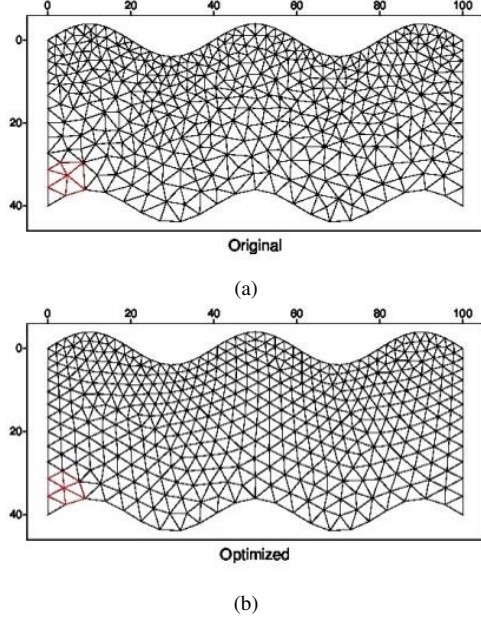

**Figure 3.** The comparison between (a) the original mesh by Delaunay Triangulation and (b) the optimised mesh by DT-CVT algorithm. Comparing with (a), the cells in (b) approach more to normal triangles. Note that both meshes have the same number of elements, and the mesher preserves the interfaces well during mesh generation.

mesh based modeling method with the finite difference algorithm in the context of wavefield simulation across strong a strong velocity contrast. The model composed of two velocity blocks is shown in Fig. 4a. We first simulate the wavefield propagation using the finite difference method. With a strong displaying scale, we can observe the artificial scattering waves caused by the staircase approximation. The strength of the artifacts depends on the ratio between the central frequency of the source function and the spatial discretization. Usually the lower the ratio is, the more likely the staircase caused artifacts will occur. As a comparison, we perform the adaptive unstructured mesh based wavefield simulation. Fig. 4c shows the generated mesh by our algorithm. While the interface is delineated well, the mesh size is adaptive to the local velocities. Fig. 4d shows the wavefield running on the generated mesh, which has no scattering artifacts caused by the staircase approximation. Although the amplitude magnitude of these artifacts are not strong, we can avoid them using our method in seismic modeling for the sake of accuracy.

We use a simplified Foothill velocity model to test the capability of our mesher in the context of rugged surface. In Fig. 5, the velocity below the rugged topography ranges from $2500\ m/s$ to $5000\ m/s$, increasing gradually with depth. There are 39 constraint points in black dots to characterize the rugged topography. We first discretize the topography according to the local velocities, with the control points being fixed. Then, we generate the unstructured mesh using the DT-CVT algorithm. We can see that while conformal to the rugged topography, the grids are adaptive to local velocities. Note that Fig. 5 is only for a quick demonstration. In numerical simulations, e.g. in Fig. 9, our mesh will be more refined and of better quality.





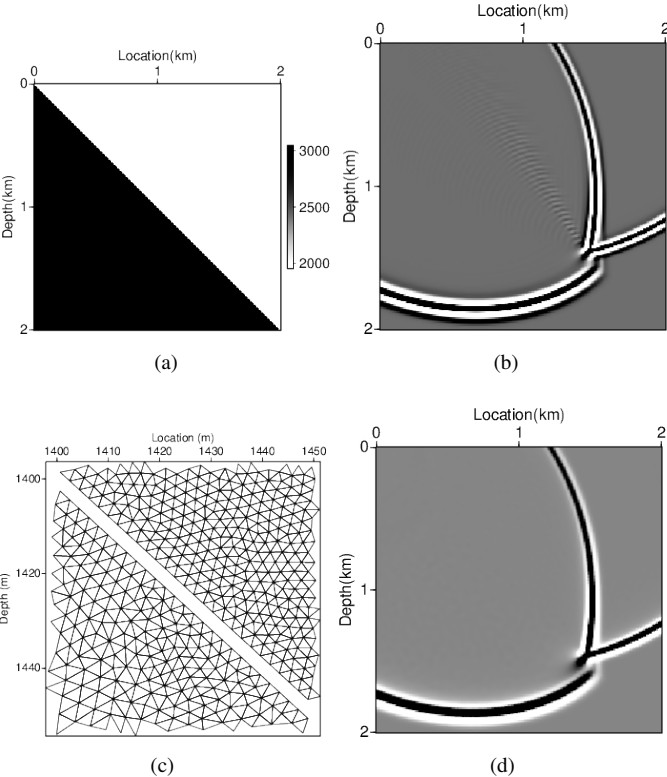

**Figure 4.** Comparison between the unstructured mesh and finite difference based wavefield simulations. (a) Two block velocity model with a tilt interface. (b) Finite-difference based wavefield simulation. We have a strong displaying scale for it to intentionally show the staircase caused scattering artifacts. (c) Adaptive unstructured mesh generated in our method. The interface is kept well without staircase approximation. (d) Wavefield simulation on the generated mesh. No scattering artifacts occur around the tilt interface.

## 2.3 Topography LSRTM workflow

When the mesher and solver are ready, we start to run the LSRTM workflow. Generally, LSRTM consists of three circular sections: (i) demigration (Born modeling), (ii) migration (RTM) and (iii) linearized inversion. Both the demigration and migration sections are based on two-way wave-equation. In $Appendix\,A$, we derive in detail the demigration and migration sections. We
5  note that both the first equation in Eq. (A-3) and the equation in Eq. (A-6) can run with Eq. (5) directly. The computation of Eq. (A-3) involves the point-source term $s\,(t; \mathbf{x}_S)$. We prepare it with $s\,(t; \mathbf{x}_S)\,M_k\,(\mathbf{x}_S)$ for Eq. (5). To run the second equation in Eq. (A-3), we need some small modifications to Eq. (5) as follows

$$M_k \left( \frac{\partial^2 (\delta P)}{\partial t^2} \right)_k = \sum_{l=1}^m (D_x (\delta P))_l (b_k)_l - \sum_{l=1}^m (D_z (\delta P))_l (a_k)_l + M_k \left( \frac{\partial^2 P_0}{\partial t^2} \right)_k \widetilde{m}_k. \tag{10}$$



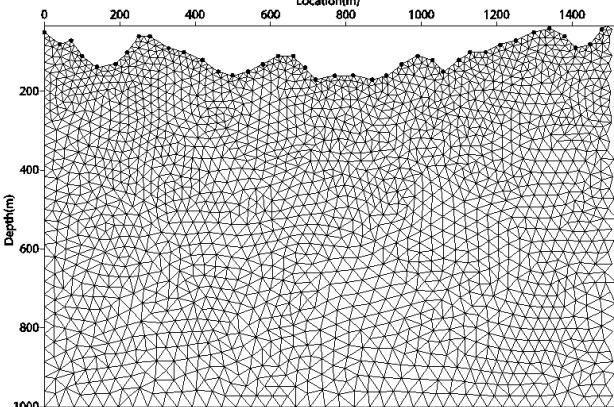

**Figure 5.** Unstructured mesh generated by DT-CVT algorithm. The 31 dark spots on the surface control the topography character. As the depth increases, the velocity model becomes faster. While conformal to the topography, the cells are adaptive to the local velocities. Note that this mesh is just for demonstration. In numerical applications, e.g. Fig. 9, we have a refined mesh, which is of better quality.

where $\widetilde{m}_k$ denotes the local reflectivity model at node $k$. In Appendix A, we express the Born modeling in a compact matrix form as

$$\mathbf{Lm} = \mathbf{d}, \tag{11}$$

with $\mathbf{L}$ being the demigration operator denoted in Eq. (A-3), $\mathbf{m}$ the logarithmic reflectivity model (Operto et al., 2000; Tromp
et al., 2005) and $\mathbf{d}$ the demigrated data. When $\mathbf{m}$ is the true reflectivity, $\mathbf{d}$ can be the observed data. In practice, the RTM profile is inferior to the true reflectivity model due to the imperfectness of the adjoint operator. To overcome its imperfectness, we attempt to approach an inverted reflectivity model via a least-squares inversion with the following $L2$ misfit:

$$E\left(\mathbf{m}\right) = \frac{1}{2}\left\|\mathbf{Lm} - \mathbf{d}\right\|_2^2 + \frac{1}{2}\lambda\left\|\mathbf{Wm}\right\|_2^2. \tag{12}$$

The regularization term in Eq. (12) is to prevent overfitting when solving an ill-posed problem. The scaling factor, which is
used to balance the importance of the regularization versus the data misfit, can be obtained by trial and error. The term $\mathbf{W}$ is to penalize roughness in the solution. LSRTM chooses to seek the inverted image via linearized inversion by using the gradient optimisation methods, such as the conjugate gradient algorithm. As stated in Appendix A, we can use the adjoint method to obtain the gradient as

$$\mathbf{m_{mig}} = \mathbf{L}^T\mathbf{d}, \tag{13}$$

Note that we are using the adjoint operator derived on a continuous level. We refer to Xu and Sacchi (2017) to provide a detailed derivation about the exact adjoint operator of the discretised system in the prestack case. Because unstructured mesh based seismic modeling involves an irregular stencil, which is not easy to seek the exact adjoint operator as that in the finite



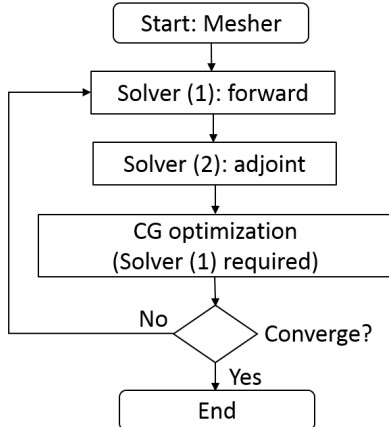

**Figure 6.** Topography LSRTM workflow. Because the background velocity in LSRTM stays the same, we run the $Mesher$ once for all. $Solver$ (1) denotes that when the input control paramter is "1", the executable runs as the forward operator. $Solver$ (2) denotes that when the input control paramter is "2", the executable runs as the adjoint operator. Note that in the $CG\ optimisation$ part, we run "1" in the $Solver$ when we need to determine the line-search step length using the conjugate gradient algorithm.

difference stencil, we leave it for our future research. It is strongly recommended to use the exact adjoint operator. By doing so, the pair of migration/demigration operators will be symmetric, which is a prerequisite for conjugate gradient method to have a better convergence behavior in LSRTM Xu and Sacchi (2017).

We design the topography LSRTM workflow following the "SPECFEM" Komatitsch and Tromp (1999) style. Fig. 6 shows
the flow chart. Because the background velocity stays the same throughout LSRTM, we run the $Mesher$ once for all. For the $Solver$ part, we give it two control options: "1" for the forward operator and "2" for the adjoint operator. To invert for the reflectivity model, we employ the conjugate gradient algorithm as the optimisation method, in which we need one forward operator and one adjoint operator to seek the line search step length (Dai and Schuster, 2013). The computational cost of iterative LSRTM is expensive, so it is necessary to run with MPI in parallel. The work is distributed as embarrassingly parallel
tasks, with one core per task. The $CG\ optimisation$ script acts as a coordinator to communicate with the $Solver$ to distribute the forward/adjoint operators over different nodes.

## 3    Numerical example

We test our methods on the Foothill model (Gray and Marfurt, 1995) in Fig. 7, which has the rugged topography with a rapid variation of elevations. The model contains some steep faults and dips, typical of mountainous onverthrust regions. The total
relief of the rugged topography is approximately $1.4\ km$. To avoid inverse crime, we use the true velocity model rather than the reflectivity model to generate the observed data. Because the free-surface multiples take no part in the conventional LSRTM, we impose no free surface onto the rugged topography. We have 100 shots spacing at $200\ m$. The source function is a $25\ Hz$





Ricker wavelet. The recording duration is $5\ s$. Each common-shot gather (CSG) has 950 receivers. The data are sampled at $20\ m$ along offset and $2\ ms$ along time. We conduct the numerical experiments on a cluster mainly with Intel Ivybridge Processor through SLURM scheduling job script.

Before the modeling stage, we prepare the rugged topography as constraints for mesh generation. The topography in Fig. 8 has 6284 reference points. We generate an unstructured mesh using the DT-CVT algorithm (Du and Gunzburger, 2002) based upon the background velocity model, which is obtained by smoothing the true velocity model with a 2D Gaussian smoother to blur the high-wavenumber details. Unlike full-waveform inversion, LSRTM keeps the same background velocity model throughout the iterative inversion. Once the mesh is ready, we can use it repetitively. The generated mesh is composed of 7325682 nodes and 14638537 cells. Even with 6284 reference points, our mesh honors each detail of the rugged topography, as shown in Fig. 9a. The local mesh varies with the local velocity, as indicated by the comparison between Figs. 9b and 9a. We can see that the grids in higher velocities are larger, and vice versa. The grids together with their local velocities determine the time step in seismic modeling. As a result, given a seismic modeling method, a mesh of adaptive cells allows for a lower computational cost compared with a mesh of fixed cells. The actual time step for this numerical example is $0.667\ ms$, which is determined by both the CFL condition and the $2\ ms$ sampling interval.

When the meshing is done, we start to run the LSRTM workflow. The migration part is using Eq. (5) and the demigration part Eq. (10). Based on the generated unstructured mesh, the sources and receivers are put onto their exact locations. When representing a point source, we use Eq. (5) with an additional source term on its right-hand side as $s\left(t; \mathbf{x}_S\right) M_{\mathbf{x}_S}$, with $s\left(t; \mathbf{x}_S\right)$ being the source function at source location $\mathbf{x}_S$. For a typical original observed data in in Fig. 10a, we cut its direct arrivals using two soft-edged windows with slopes of +/- $0.25\ s/km$, resulting in Fig. 10b. We first forward-propagate the source wavefield, and store the wavefield histories at the inner PML boundaries together with the last two wavefield frames. These stored information will be used for source wavefield reconstruction in RTM. During the RTM stage, by using the adjoint method, we inject the seismic data through the receivers on the topography, and cross-correlate the backward propagated receiver wavefield with the forward propagated source wavefield, with redundant source effects considered (Liu et al., 2016). Fig. 12a shows the RTM image. Then, we run the demigration operator to get the predicted data. The data residual is the difference between the observed and predicted data. We seek the inverted image via the conjugate gradient method (Dai and Schuster, 2013). As LSRTM runs, the data misfit reduces, as shown in Fig. 11. Although here the LSRTM converges, a better convergence behavior would be expected with the development of the exact discretized adjoint operator instead of the continuous one. Taking the $35th$ CSG for example, we can see that after 20 iterations, the predicted data in Fig. 10c matches well with the observed data in Fig. 10b, with their data residual in Fig. 10d. Fig. 12b shows the inverted image. Fig. 13 shows the zoom-in views of the true reflectivity, the RTM and LSRTM images for detailed investigations. With Fig. 13a for reference, yy comparing Figs. 13b and 13c, we can see that in the case of rugged topography, the LSRTM image has more balanced amplitude distribution and higher imaging resolution than the RTM image, especially at the steep dips and faults.



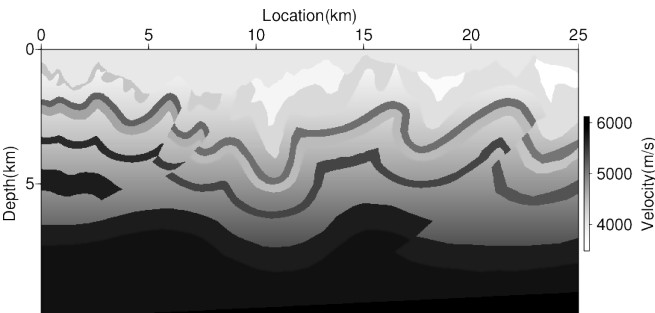

**Figure 7.** The Foothill velocity model with rugged topography.

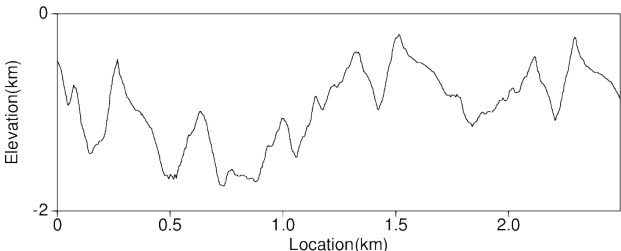

**Figure 8.** Rugged topography of the Foothill model.

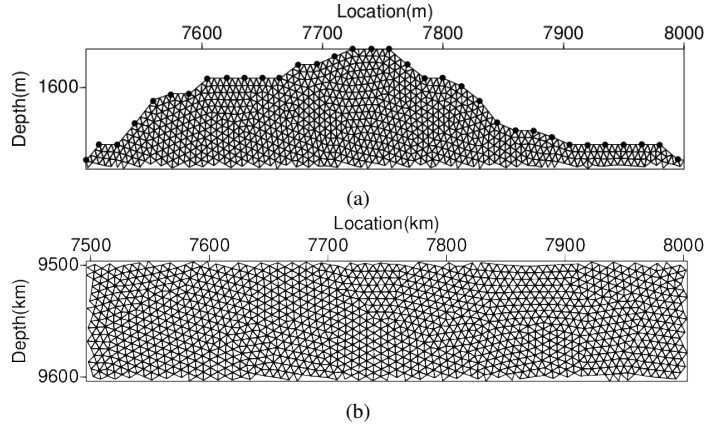

**Figure 9.** Local meshes of (a) near-surface and (b) deep parts generated by DT-CVT for the Foothill velocity model. We can see that the mesh in (a) honors the near-surface details (as shown by the black topography control points). Also, from a panoramic view, we can see that the cells in (b) are sparser than (a), meaning that the grids in (b) are larger than (a) thanks to adaptive meshing.

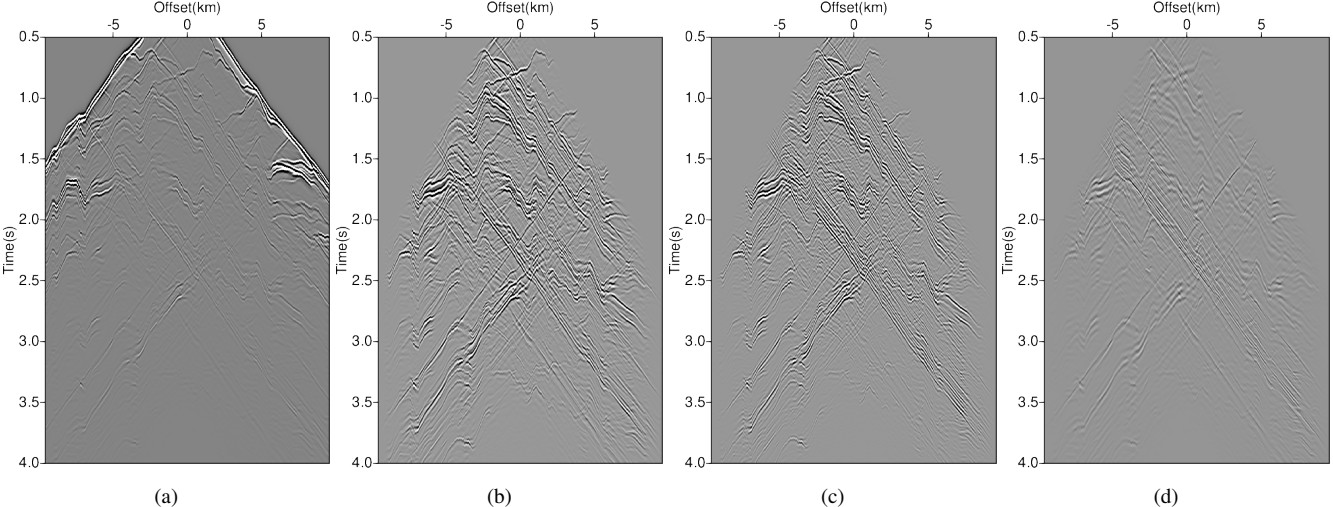

**Figure 10.** CSGs of the $35th$ shot. (a) and (b) the observed data before and after muting direct arrivals by soft-edged windows of slope +/-
0.25 $s/km$; (c) the demigrated data after 20 iterations; (d) the residual between (b) and (c). Note that (b), (c), and (d) are shown with the
same colour scale. In (d) there remain many coherent events in the residual. For the events of lower frequency bands, we explain about them
as a consequence that LSRTM is a high-wavenumber reflectivity inversion. For the events of similar frequency bands, we explain about them
as a result of the rugged topography scattering. The topography reflectivity cannot be updated well during LSRTM due to the presence of the
source points on the topography.

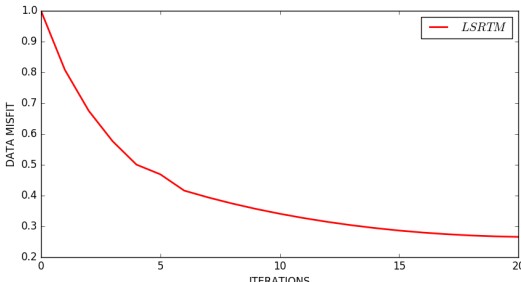

**Figure 11.** The normalized data-misfit curve of topography LSRTM, which shows that our LSRTM converges.





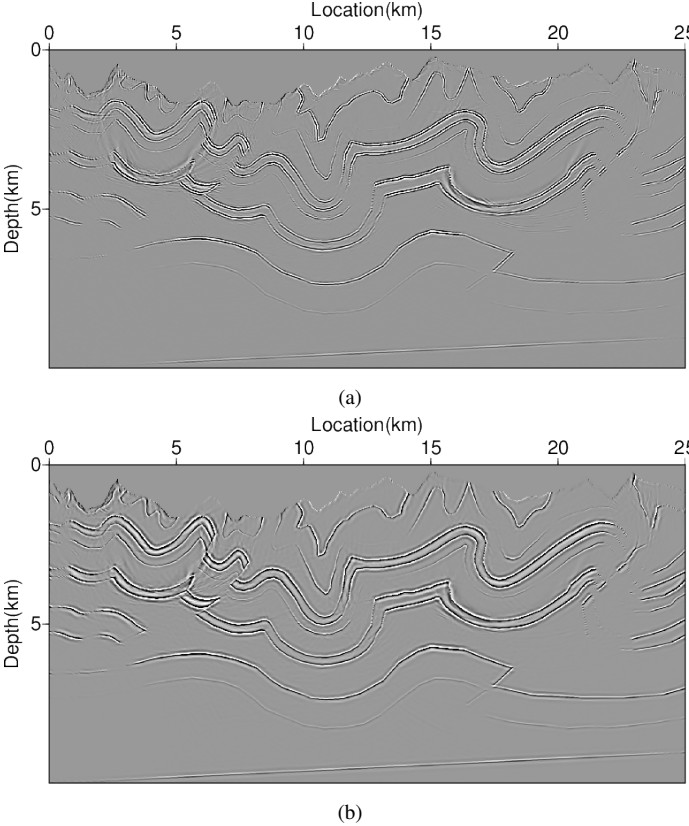

**Figure 12.** In the case of rugged topography, (a) RTM and (b) LSRTM images. We can see that (b) has better amplitude distribution and higher imaging resolution than (a), especially for the steep dips and faults.

## 4 Conclusions

We present an unstructured mesh-based workflow which can handle the LSRTM from rugged topography. Our approach is free of static correction. It mainly consists of three principal parts: mesher, solver, and LSRTM. We generate the unstructured mesh with the DT-CVT algorithm, with the rugged topography serving as boundary constraints. Our mesher honors details on the topography without approximations such as flattening and so on. Also, provided a modeling method, a mesh of adaptive cells allows a lower computational cost compared with a mesh of fixed cells. The solver to simulate the wavefield propagation is an unstructured mesh-based modeling scheme named the grid method. The LSRTM is in its conventional style. In this workflow, we do not need to do static correction before imaging and inversion, but directly inject wavefield downwards through the receivers on the rugged topography. A test on the Foothill model validate our scheme. In comparison with the conventional RTM from rugged topography, the LSRTM from rugged topography promises an imaging profile with more balanced amplitude distribution and higher resolution, especially for the steep dips and faults.




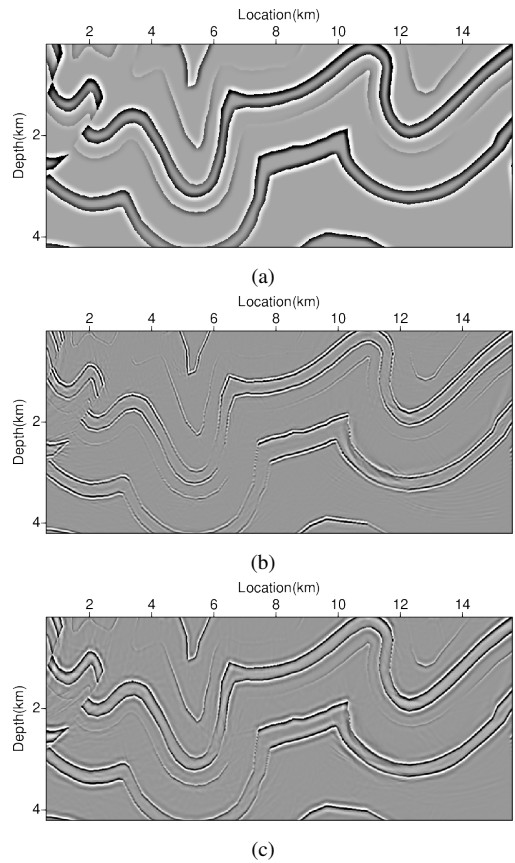

**Figure 13.** Zoom-in comparison views for detailed investigations. (a) Part of true reflectivity. (b) Part of the topography RTM image. (c) Part of the topography LSRTM image. We see that in reference to (a), both (b) and (c) have the reflectors in the right places, but the image in (c) looks sharper with more balanced amplitude distributions.

## Appendix A: Parameterisation in LSRTM

The time-domain acoustic wave-equation reads

$$\left(\frac{1}{c(\mathbf{x})^2}\frac{\partial^2}{\partial t^2} - \nabla^2\right)P(\mathbf{x}, t; \mathbf{x}_S) = s(t; \mathbf{x}_S), \tag{A1}$$

where $c(\mathbf{x})$ is the true velocity model, $P(\mathbf{x}, t; \mathbf{x}_S)$ the pressure wavefield, $s(t; \mathbf{x}_S)$ the shot at $\mathbf{x}_S$. We have $c(\mathbf{x}) = c_0(\mathbf{x}) +$
5 $\delta c(\mathbf{x})$, with $c_0(\mathbf{x})$ being the background velocity and $\delta c(\mathbf{x})$ the velocity perturbation. $c_0(\mathbf{x})$ accounts for the traveltime information, and $\delta c(\mathbf{x})$ the reflection waves. Expanding the perturbation term $1\big/(c_0(\mathbf{x}) + \delta c(\mathbf{x}))^2$ around $c_0(\mathbf{x})$ into a Taylor series and neglecting the quadratic term, we can get

$$\frac{1}{(c_0(\mathbf{x}) + \delta c(\mathbf{x}))^2} \approx \frac{1}{c_0(\mathbf{x})^2} - \frac{2\delta c(\mathbf{x})}{c_0(\mathbf{x})^3}. \tag{A2}$$




Substituting Eq. (A-2) into Eq. (A-1) yields

$$
\begin{cases}
\left(\frac{1}{c_0(\mathbf{x})^2}\frac{\partial^2}{\partial t^2}-\nabla^2\right)P_0\left(\mathbf{x},t;\mathbf{x}_S\right)=s\left(t;\mathbf{x}_S\right), \\
\left(\frac{1}{c_0(\mathbf{x})^2}\frac{\partial^2}{\partial t^2}-\nabla^2\right)\delta P\left(\mathbf{x},t;\mathbf{x}_S\right)=\frac{m(\mathbf{x})}{c_0(\mathbf{x})^2}\frac{\partial^2 P_0(\mathbf{x},t;\mathbf{x}_S)}{\partial t^2},
\end{cases}
\tag{A3}
$$

in which we parameterise the logarithmic reflectivity as $m\left(\mathbf{x}\right)=2\delta c\left(\mathbf{x}\right)/c_0\left(\mathbf{x}\right)$. There are two wave-equations in Eq. (A-3). The first one is the zero-order wave-equation propagating the background wavefield; the second one is the first order wave-equation scattering reflected wavefields. Two modelings are required: one for $P_0$ and the other for $\delta P$. For the sake of simplification, we represent the demigration system in Eq. (A-3) in a compact matrix form:

$$
\mathbf{Lm}=\mathbf{d},
\tag{A4}
$$

with $\mathbf{L}$ being the forward operator, $\mathbf{m}$ the reflectivity model, and $\mathbf{d}$ the data. Following the adjoint method Tromp et al. (2005), we apply the following imaging condition to obtain the gradient:

$$
m_{\delta c}\left(\mathbf{x}\right)=\int_0^T-\frac{\partial^2 P_0}{c_0^2\partial t^2}\widehat{P}dt,
\tag{A5}
$$

with $P_0$ being the forward-propagated source wavefield obtained from Eq. (A-3), and $\widehat{P}$ the backward-propagated adjoint wavefield governed by

$$
\left(\frac{\partial^2}{c_0(\mathbf{x})^2\partial t^2}-\nabla^2\right)\widehat{P}\left(\mathbf{x},t;\mathbf{x}_S\right)=\Delta d(\mathbf{x}_R,t;\mathbf{x}_S),
\tag{A6}
$$

with $\Delta d(\mathbf{x}_R,t;\mathbf{x}_S)$ being the residual between observed and predicted data. Note that we preprocess $\widehat{P}$ with the second-order time derivative. We also represent the migration system in Eqs. (A-5) and (A-6) in a compact matrix form:

$$
\mathbf{m_{mig}}=\mathbf{L}^T\mathbf{d},
\tag{A7}
$$

with $\mathbf{L}^T$ being the adjoint operator, $\mathbf{d}$ the seismic data, and $\mathbf{m_{mig}}$ the imaging profile. Provide more details. We solve Eq. (A-6) by using a discretisation of the continuous adjoint. The adjoint of the discretised system will be our next move because it provides a better convergence behavior in LSRTM (Xu and Sacchi, 2017). Also, we refer to Dai and Schuster (2013) for a more complete derivation of the LSRTM parameterisation and equations.

*Acknowledgements.* We would like to thank H. Gao for fruitful discussions. We would like to thank D. Keyes and D. Peter's supporting for the High-Performance Computing facilities. The research reported in this publication was supported by funding from King Abdullah University of Science and Technology (KAUST). For computer time, this research used the resources of the Information Technology Division and Extreme Computing Research Center (ECRC) at KAUST.




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
