# Peer review of "An adaptive unstructured mesh based solution to topography least-squares reverse-time imaging"

_Solid Earth, 2019_

## Referee Comment (RC1) · Tristan van Leeuwen (Referee) · 22 Mar 2019

I have previously reviewed this manuscript as it was submitted to a different journal. The paper was rejected because of the lack of novelty; many of the ingredients in the paper have been described in earlier work by the same authors. In particular, there is a significant overlap with this paper: https://doi.org/10.1111/1365-2478.12415. The authors spend one sentence (page 2 line 7) on this work. More elaborate discussion of the contributions of this work with respect to previous work is needed.

---

## Referee Comment (RC2) · Anonymous Referee #2 · 14 Apr 2019

This paper describes a LSRTM scheme in the presence of topography, which is an extension of the authors' previous work on RTM. This is reasonable, but the authors do not talk about the specific issues of LSRTM. In the current literature, there are several researchers already dealing with topography with LSRTM, but the authors did not talk any of them. What is the advantage of this method to the current literature.

Another issue I am concerning is that, wave propagation in the presence of topography is very complex. Dealing with this problem on the acoustic approximation is a big issue. No surface waves are simulated.

Since this paper is trying to dealing with practical problem, I hope the authors can

present it in a more practical way.
* * *